# Sources and Drivers of ARGs in Urban Streams in Atlanta, Georgia, USA

**DOI:** 10.3390/microorganisms10091804

**Published:** 2022-09-08

**Authors:** Robert A. Sowah, Marirosa Molina, Ourania Georgacopoulos, Blake Snyder, Mike Cyterski

**Affiliations:** 1Oak Ridge Institute for Science and Education, Oak Ridge, TN 37831, USA; 2U.S. EPA, Laboratory of Services & Applied Sciences Division, 980 College Station Rd., Athens, GA 30605, USA; 3U.S. EPA, Office of Research and Development, Center for Environmental Measurement and Modeling, 109 T. W. Alexander Drive, Durham, NC 27709, USA; 4Student Services Contractor to the U.S. EPA, Office of Research and Development, Center for Environmental Measurement and Modeling, 960 College Station Rd., Athens, GA 30605, USA; 5U.S. EPA, Office of Research and Development, Center for Environmental Measurement and Modeling, 960 College Station Rd., Athens, GA 30605, USA

**Keywords:** antibiotic resistance, urban streams, waterborne pathogens, fecal indicator bacteria

## Abstract

The spread of antibiotic resistance genes (ARGs) in the aquatic environment is an emerging concern in the interest of protecting public health. Stemming the environmental dissemination of ARGs will require a better understanding of the sources and drivers of ARGs in the water environment. In this study, we used direct measurement of sewage-associated molecular markers, the class 1 integron gene, standard water quality parameters, and watershed characteristics to evaluate the sources and drivers of ARGs in an urban watershed impacted by a gradient of human activities. Quantitative polymerase chain reaction (qPCR) was used to quantify the abundance of the sewage-associated HF183, the *E*. *coli* fecal indicator, class 1 integron gene (*int*1), and the ARGs *sulI*, *sulII*, *tet*W, *tet*M, *ampC,* and *bla*SHV in stream water samples collected from the Proctor Creek watershed in Atlanta, Georgia. Our findings show that ARGs were widely distributed, with detection frequencies of 96% (*sulI* and *sulII*), 82% (*tet*W and *tet*M), and 49% (*ampC* and *bla*SHV). All the ARGs were positively and significantly correlated (*r* > 0.5) with the HF183 and *E*. *coli* markers. Non-linear machine learning models developed using generalized boosting show that more than 70% of the variation in ARG loads in the watershed could be explained by fecal source loading, with other factors such as class 1 integron, which is associated with acquired antibiotic resistance, and environmental factors contributing < 30% to ARG variation. These results suggest that input from fecal sources is a more critical driver of ARG dissemination than environmental stressors or horizontal gene transfer in aquatic environments highly impacted by anthropogenic pollution. Finally, our results provide local watershed managers and stakeholders with information to mitigate the burden of ARGs and fecal bacteria in urban streams.

## 1. Introduction

Antibiotic resistance genes (ARGs) and genetic determinants are increasingly being recognized as contaminants in the aquatic environment [1,2,3,4,5]. The U.S. Environmental Protection Agency (U.S. EPA), in its 5-year review of the 2012 recreational water quality criteria, cites ARGs and antimicrobial resistant bacteria (ARB) as emerging concerns in the interest of protecting public health [6]. The presence of ARGs and ARB in the aquatic environment is gaining attention as a critical water quality and public health concern due to research linking environmental and clinical resistance [3,7]. It has also been suggested in recent studies that the rise in community-acquired ARB infections can be attributed to the emergence and spread of ARGs and ARB in the environment [8,9,10,11]. The development of methods and tools to understand the contribution of the environment to the spread of ARGs and ARB is a critical step to addressing the global antibiotic resistance menace [12,13] and supports the One Health approach, which recognizes the connection between the health of people, animals, and the environment. 

Although some ARGs occur naturally in the environment, the rapid evolution and spread of new ARGs have been attributed to human activities including the widespread use or misuse of antibiotics for human and animal purposes [14,15,16]. In general, three main factors are recognized for driving the spread of ARGs in the aquatic environment. These include wastewater/sewage discharges from human or animal sources, selection pressure exerted by antibiotics, metals, and other contaminants in the water environment, and horizontal gene transfer between microorganisms in the environment. Much work has been done to characterize ARG loads at the source level, however, questions remain about the contributions of these sources/drivers to ARG loads in each system.

The role of the aquatic environment in the spread of ARGs and ARB cannot be overemphasized, as the water environment provides a pathway for contact with resistance determinants and pathogens through recreational water use, the consumption of crop products irrigated with contaminated water, and the consumption of meat products from animals that have come into contact with contaminated surface waters [16,17,18]. Urban streams, by nature of the impact of anthropogenic activities, are subject to multiple stressors that can promote the emergence and spread of resistance determinants in the environment. Traditionally, urban waters have been known to be impacted by nutrients, pharmaceuticals, metals, and pathogens through multiple stressors such as wastewater discharges, stormwater run-off from large areas of impervious cover, and old and/or failing sewage infrastructure that often results in sewage overflows to the aquatic environment. The combined effect of these anthropogenic sources has been known to increase the loading of ARGs in urban streams, above what is naturally observed in the environment [5,15,19]. Accordingly, Sanderson et al. [1] opined that these polluted waters have become reservoirs of ARGs and a conduit for the dissemination of antibiotic resistance genes and determinants to new environments. The aquatic environment, moreover, provides an interface for the exchange of ARGs between pathogens and environmental bacteria, which can lead to the development of new resistance mechanisms [20].

Recent studies have shown that the direct measurement of sewage-associated microbial molecular markers can be used to assess the sources and drivers of ARGs in the aquatic environment. Ahmed et al. [21] demonstrated the efficacy of this approach using the sewage-associated HF183 *Bacteroidales* marker and the *crAssphage* bacteriophage marker to identify the sources of ARGs in storm drain outfalls off the Florida coast. Metadata analysis by Karkman et al. [22] used the *crAssphage* marker to understand the connection between fecal pollution and ARG patterns in sewage-polluted environments. The authors observed that, in sewage-polluted environments, the presence of ARGs is largely explained by fecal pollution, whilst the effect of factors such as selective pressure was only evident in environments polluted by very high levels of antibiotics from manufacturing. 

Although these studies provide insight into the dynamics of ARGs in the aquatic environment, the narrow scope of these studies, in terms of the sources considered (storm drain outfalls) or the indicators used, do not allow for a detailed analysis of sources and drivers of ARG abundance and spread in the environment. This is particularly important in urban watersheds such as the Proctor Creek watershed in Atlanta, GA, USA, which has a long history of water quality impairment from multiple sources. Anthropogenic impacts in this watershed stem from sanitary sewer overflows, industrial activities, and runoff from highly developed urban centers and associated impervious areas [23,24,25]. A study by the U.S. EPA showed that despite the progress made in the watershed, including the decommissioning of combined sewer systems, the Proctor Creek watershed still suffers from elevated concentrations of fecal contaminants and nutrients in streams [6]. The goal of this study was to demonstrate a watershed-level approach to identifying the sources and drivers of ARGs in the Proctor Creek watershed. In this study, we investigated the role of human fecal pollution, water quality parameters, and watershed characteristics in the abundance and spread of ARGs in urban streams. Predictive models were developed to quantify the influence of these factors on ARG concentrations in the aquatic environment. 

## 2. Materials and Methods

### 2.1. Study Area

The study was conducted in the Proctor Creek watershed located within the City of Atlanta, GA, USA (Figure 1). The Proctor Creek watershed drains the west side of the city of Atlanta, its headwaters starting near the city center and flowing northwest for approximately 15 km into the Chattahoochee River (a major water body for recreation, agricultural, and fishing purposes in the tri-states of Georgia, Florida, and Alabama). The watershed drains an area of ~41 km^2^. The area around the headwaters is highly impervious, with most of the streams either piped underground or channelized aboveground [6]. The headwaters receive urban runoff from the downtown Atlanta metropolitan area and the rest of the watershed is impacted by multiple types of point and nonpoint sources of pollution including combined sewer overflow (CSO) facilities, railroads, a large freight yard along the northern boundary of the watershed, landfills, automotive salvage yards, and illegal trash dumps [6]. 

### 2.2. Sample Collection

Grab surface water samples were collected in 1 L sterile Nalgene bottles on a bi-weekly basis from 12 sampling locations distributed across the Proctor Creek watershed. The sampling points consist of 6 sites that are located on the mainstem of the creek and 6 tributary sites (Figure 1). Samples were kept on ice during transport to the U.S. EPA’s Office of Research and Development (ORD) laboratory in Athens, GA, USA and processed within 6 h of sample collection following guidelines in the National Field Manual for the Collection of Water Quality Data [26]. Samples (*n* = 287) were collected over a 1-year period from May 2016 to April 2017. During sampling, in situ water quality parameters, including dissolved oxygen (DO), specific conductance (SPC), pH, and temperature, were measured using a multiparameter probe (YSI, Yellow Springs, OH, USA). 

### 2.3. Water Quality Analysis

Water samples were analyzed for culturable *Escherichia coli* (*E*. *coli*) using the Colilert-defined substrate method (IDEXX Laboratories Inc., Westbrook, ME, USA). Samples were diluted 10-fold and 100-fold in 100 mL volumes and run in duplicate to obtain average *E*. *coli* concentrations in MPN/100 mL. Additionally, turbidity measurements were performed on water samples in the laboratory using a Turbidimeter (LaMotte Company, Chestertown, MD, USA).

Fifty-milliliter samples were filtered in quadruplicate on 0.45 µm polycarbonate filter membranes (Millipore Sigma, Burlington, MA, USA) to concentrate microbial DNA for molecular analysis. Filters with microbial residue were placed into 2 mL microcentrifuge tubes and stored in a freezer at −80 °C until further processing. Two filters were subsequently extracted using the DNeasy PowerLyzer PowerSoil kit (Qiagen, Germantown, MD, USA) and recommended protocol with some modifications (see Appendix A for details). The extracted DNA was then stored at −80 °C until further analysis. The two remaining filters were kept in storage as a backup. The extracted DNA was analyzed for fecal indicator markers EPA EC 23S and HF183/BacR287, which will be referred to as *E*. *coli* marker and HF183, respectively, going forward. Other genetic markers that were analyzed include the 16S rRNA, which measures microbial community abundance; class 1 integron gene (*int1*), which is associated with acquired antibiotic resistance; and the sulfonamide resistance genes (*sulI*, *sulII)*, tetracycline resistance genes (*tet*W, *tet*M), and beta-lactam resistance genes (*amp*C and *bla*SHV). Among all the ARGs monitored in the aquatic environment in previous studies, the sulfonamide and tetracycline resistance genes are the most abundant and widely distributed, followed by beta-lactam genes [27]. All the quantitative polymerase chain reaction (qPCR) assays were run on a QuantStudio 3 instrument. All qPCR assays in this study followed reaction conditions described in previous studies unless stated otherwise (see Appendix A for reaction conditions). Taqman qPCR assays were run for all markers except for the *sulI*, *amp*C, and *bla*SHV markers which were run using SYBR Green chemistry. The Taqman Universal Master Mix (2X) was used in Taqman assays whilst the PowerUp SYBR Green Master Mix (2X) was used for SYBR assays in a final reaction volume of 20 µL. Marker quantification was performed using standard curves that were generated during each qPCR run. The Appendix A provides additional details on the qPCR procedures and the quality control protocols implemented. 

### 2.4. Data Analysis

The qPCR and culturable *E*. *coli* data were log-transformed for statistical analysis. Samples that were non-detectable for culturable *E*. *coli* were given a log-transformed value of 0 (i.e., the detection limit of 1 MPN/100 mL). Molecular markers that were below the lower limit of quantification (LLOQ), and thus undetected in this study, were imputed using regression on order statistics (ROS), as described in Sowah et al. [28]. The LLOQ for qPCR assays was determined as the lowest copy number of the qPCR standards that was accurately measured in all assays [29,30,31,32]. qPCR data in gene copies (GC) per ml were normalized with the copies of the 16S rRNA gene to yield the relative abundance of genetic markers in samples. All data analysis was performed in the R program [33] with significance determined at the 5% significance level. The relationships between ARGs and fecal indicators, integrase gene, standard water quality parameters, and antecedent rainfall were examined using the non-parametric Spearman correlation analysis. Principal component analysis (PCA) was performed to create an index score variable that is an optimally weighted combination of the six ARGs (*sulI*, *sulII*, *tet*M, *tet*W, *ampC,* and *bla*SHV) measured in this study. The rationale for this approach was to reduce the number of response variables (ARGs) into one or two principal components that explain most of the variability in the data and can be used as response variables in predictive modeling. The *prcomp* function in the VEGAN package in R was used for PCA. The first two principal components (referred to as PC1 and PC2) were used as the response variables in our predictive models. 

#### 2.4.1. Generalized Boosted Models

We used a Generalized Boosted Model (GBM), a machine learning regression/classification method based on constructing a chain of decision trees [34] to determine which independent variables were most influential in predicting PC1 and PC2 scores. The independent variables evaluated include *int1*, 16S rRNA, culturable *E*. *coli*, *E*. *coli* marker, HF183, SPC, DO, pH, water temperature, turbidity, antecedent rainfall (24 h, 48 h, and 72 h), and site location. We developed models for PC1 and PC2 separately using the *gbm* package (version 2.1.5, 11) in R. There is some randomness in the results of the GBM algorithm when fitting a given dataset, so we developed bootstrapped estimates of model performance measures by fitting 500 models for PC1 and PC2. In each of the 500 bootstrap iterations, the dataset was randomly split: 80% of the dataset was randomly placed into a training set, and the remaining 20% was put into a testing set. The *gbm* algorithm uses multiple hyperparameters that influence the fitting process and final solution. Hyperparameter values were set as follows: Error term = Gaussian, Maximum number of trees = 10,000, Shrinkage = 0.005, interaction depth = 3, Bag fraction = 0.5, Train fraction = 1, Number of minimum observations per node = 5, and cross-validation folds = 10. See the Appendix A for a discussion of hyperparameter selection methods. 

#### 2.4.2. Partial Dependence Plots

The *gbm* package in R allows for the creation of Partial Dependence Plots (PDPs), which graphically summarize the relationship between a single covariate and its influence on the response variable. One option within the *gbm* package will allow for maximum flexibility in the PDP, permitting it to increase or decrease amidst multiple inflection points over the range of covariate values. The modeler can, however, constrain the covariate/response relationship (via a parameter called “var.monotone”) to be monotonically increasing or decreasing, resulting in easier-to-interpret PDPs, but possibly also affecting the model’s predictive performance. To constrain model complexity (i.e., limit the number of significant covariates), we modeled PC1 using two stages: first, we fit 500 models to PC1 using all the predictors; second, we fit 500 additional simplified models using only those covariates with >3% average influence from the initial stage. Next, we examined the stage-two PDPs to see if their relationship could be adequately described/captured using a monotonically increasing/decreasing function. Finally, a third set of 500 models was run using the “var.monotone” parameter to constrain the covariate/response relationship for certain covariates. We call this the “Refined” PC1 model, and we created aggregate PDPs for each covariate in the Refined PC1 model by using the partial dependence points for each of the 500 model runs from the Refined models. The 2.5th, 50th, and 97.5th percentiles of the response values at each evaluated covariate value were taken from this array of points. These PDPs show how much the PC1 score would change (adjustment value) on the y-axis across a specific covariate’s range on the x-axis. The average PC1 adjustment values (±2 standard errors) were used to evaluate significant differences between the sampling sites.

## 3. Results

### 3.1. Concentration of 16S rRNA, ARGs, Fecal Markers, and int1

The concentration of 16S rRNA varied from 2.3 Log_10_ to 7.6 Log_10_ GC/mL with an average of 6.0 Log_10_ GC/mL of water (Figure 2). The concentrations of ARGs, fecal markers, and *int1* markers are summarized in Figure 3A. The class 1 integron gene *int1* was detected frequently (98%, *n* = 287) in the watershed. The concentration of *int1* ranged from 1.3 to 5.6 Log_10_ GC/mL with an average concentration of 3.9 Log_10_ GC/mL. The average concentrations of the *E*. *coli* and the HF183 markers were very similar with values of 2.1 Log_10_ GC/mL and 2.2 Log_10_ GC/mL, respectively. *E*. *coli* was detected in 87% (*n* = 287) of samples and concentrations varied from 0.71 to 4.0 Log_10_ GC/mL. The detection frequency of HF183 was 63% (*n* = 287) with concentration between 0.5 and 5.0 Log_10_ GC/mL. Among the ARGs, the detection frequency varied from a high of 97% for *sulI* to a low of 45% for *bla*SHV (Table 1). A comparison of the detection frequencies between the different ARGs shows that the sulfonamide resistance genes *sulI* (97%) and *sulII* (95%) were the most frequent, followed by the tetracycline resistance genes *tet*M (87%) and *tet*W (78%), and lastly, the beta-lactam resistance genes *ampC* (53%) and *bla*SHV (45%). The concentrations of ARGs also followed a similar trend as the detection frequencies. ARG concentrations were as follows: *sulI* (1.1–5.4 Log_10_ GC/mL), *sulII* (1.1–5.9 Log_10_ GC/mL), *tet*M (0.6–4.9 Log_10_ GC/mL), *tet*W (0.6–5.2 Log_10_ GC/mL), *ampC* (0.3–4.7 Log_10_ GC/mL), and *bla*SHV (0.2–4.8 Log_10_ GC/mL). 

### 3.2. Culturable E. coli and Standard Water Quality Parameters

Details of the distribution of culturable *E*. *coli* and water quality parameters including DO, SPC, pH, temperature, and turbidity are summarized in Appendix A. The data show that the culturable *E*. *coli* concentration varied from 0 to 4.6 Log_10_ MPN/100 mL with an average of 3.0 Log_10_ MPN/100 mL on the pooled data. The average temperature, SPC, pH, DO, and turbidity were 18 °C, 270 µS/cm, 7.4, 7.9 mg/L, and 8.1 NTU, respectively. 

### 3.3. Relative Abundance of ARGs, Fecal Markers, pr and int1

We determined the relative abundances of the ARGs, fecal, and *int1* markers by normalizing the gene concentrations with the concentration of the 16S rRNA gene. This gives us an idea of the abundance of ARGs, fecal, and *int1* genes in relation to the total microbial community abundance in the streams. The relative abundance of *sulI* (−2.7 Log_10_ GC/mL), *sulII* (−2.9 Log_10_ GC/mL), *tet*M (−3.7 Log_10_ GC/mL), *tet*W (−3.5 Log_10_ GC/mL), *ampC* (−4.4 Log_10_ GC/mL), and *bla*SHV (−4.6 Log_10_ GC/mL) are summarized in Figure 3B. 

### 3.4. Correlation of ARGs, Fecal Markers, 16S rRNA, int1, and Water Quality Parameters

Figure 4 shows a correlation plot that depicts the relationships between ARGs, water quality, and environmental parameters measured. Other watershed level characteristics such as the antecedent rainfall prior to sample collection were also evaluated to determine whether there is any association with ARG abundance in the watershed. The strength of the correlation was considered “strong” when the correlation coefficient *r* > 0.7, “moderate” when 0.4 ≤ *r* ≤ 0.7, and “weak” when *r* < 0.4. The same criteria were used to describe the relationships in the negative direction. The relationships between the ARGs were positive and moderate to strong, with *r* values ranging from 0.4 to 0.86. The relationships were particularly strong within the ARG groups rather than between groups. For example, a strong correlation (*r* = 0.86) was observed between *tet*M and *tet*W, *sulI* and *sulII* (*r* = 0.72), and *ampC* and *bla*SHV (*r* = 0.74). The relationships between the different ARG groups, *tet* and *sul* genes (*r* = 0.54 to 0.68), *tet* and beta-lactam genes (*r* = 0.59 to 0.7), and *sul* and beta-lactam genes (*r* = 0.40 to 0.63), were positive and varied from moderate to strong.

The fecal markers *E*. *coli* and HF183 were also positively correlated with the ARGs, with *r* ranging from 0.53 to 0.89. Both *E*. *coli* and HF183 markers were strongly correlated with the *tet* genes, moderately correlated with the *sul* genes, and a moderate and a strong correlation was observed with the beta-lactam genes. The correlation of ARGs with culturable *E*. *coli* was strong for the *tet* genes and moderate for the *sul* genes and beta-lactam genes. The *int1* gene was strongly correlated with the *sulI* gene (*r* = 0.77) and moderately correlated with *sulII* (*r* = 0.49). Weak correlations were, however, observed between *int1* and *tet* (*r* = 0.17 to 0.33) and beta-lactam genes (*r* = 0.14 to 0.17). Moreover, the 16S rRNA gene was moderately correlated with ARGs, with *r* ranging from 0.4 to 0.5. The relationships between ARGs and water quality parameters including DO, pH, turbidity, temperature, and SPC were weak, whilst antecedent rainfall was generally weakly correlated with ARGs except for *tet*M, which was moderately correlated (*r* = 0.40) with 24 h antecedent rainfall. 

### 3.5. Predictive Modeling with Generalized Boosted Models

We developed predictive models with a generalized boosted machine learning algorithm using the PC1 and PC2 scores as the response variables. The results of the PCA (Appendix A) show that 82% of the variation in the ARGs could be explained by the first two principal components, with PC1 contributing 70% to the total variation in the data. The performance of the *gbm* package on training and test data is provided in Table 2. For each of these models, the average R^2^ values for the testing data were about 12% lower than the average R^2^ values for the training data, indicating some, but not severe, model overfitting. In addition, the Refined PC1 saw only a small decrease (0.04 R^2^ units) in performance after removing less influential covariates and constraining some of the remaining covariates. Table 3 shows the average influence of each predictor in the PC1 model, the Refined PC1 model, and the PC2 model. 

As stated earlier, the covariates with an average influence >3% in the PC1 model were retained and used to develop the Refined PC1 model. The results from the initial model runs for PC1 and PC2 shows that the *E*. *coli* marker, HF183, site, *int1*, culturable *E*. *coli,* and water temperature were the most significant in explaining the variation in ARGs in the watershed, contributing 88% and 66% of the average influence for PC1 and PC2, respectively. The refined PC1 model shows that the *E*. *coli* marker and HF183 together explain a significant proportion (66%) of the variability in ARGs. Site was the next influential, followed by the *int1* gene, water temperature, culturable *E*. *coli,* and 16S rRNA. We present PDPs showing the average influence of the predictors on PC1 (analysis limited to PC1, which explained a significant proportion of the variance in ARGs) in Figure 5. The PDPs for the *E*. *coli* marker and the HF183 show that as the marker copies increase, the adjustment to PC1 scores becomes less negative, meaning an increase in ARG concentration. Culturable *E*. *coli*, *int1*, and 16S rRNA all had a positive influence on ARG abundance; water temperature showed an interesting relationship (Figure 5f). PC1 scores were high at low temperatures, then decreased at temperatures from 6.4 °C to ~20 °C, then increased again for temperatures above ~20 °C. Figure 6 shows the average influence of the sampling sites on the PC1 scores and ARGs abundance. The results show that GR had by far the most ARG abundance in the watershed, followed by LST, NAVE and HT, and NAVECSO. The sites with the lowest PC1 scores (and thus ARG abundance) were NW and LC, followed by JJ, KC, and SST. 

## 4. Discussion

The contribution of urban watersheds to the emergence and spread of ARGs is now being fully recognized as part of the One Health approach to combatting antibiotic resistance. Urban streams such as Proctor Creek can act as reservoirs for the emergence and spread of ARGs in the environment. Our results show that ARGs occur frequently in streams in the Proctor Creek watershed over the study period. The high prevalence and abundance of *sul* genes in this study are comparable to findings from previous studies [21,27,35]. Ahmed et al. [21] reported elevated levels of *sul* genes in storm drain outfalls in Tampa, FL, USA during dry and wet conditions and concluded that fecal pollution was the most likely source of the ARGs. *S**ul* genes were also found to be highly prevalent and abundant in the Huangpu River watershed in China with average concentrations four and five orders of magnitude higher than *tet* and beta-lactam genes, respectively [27]. The high prevalence of *sulI* and *sulII* genes is not surprising given the widespread use of sulfonamide antibiotics, the high abundance of these genes in wastewater, and their association with mobile genetic elements [27,36].

At the watershed level, ARGs may originate from animal sources, sewage or wastewater discharges, selective pressure due to the presence of selective agents such as antibiotics and metals, or may occur naturally in the environment. Although ARG burden from sources such as wastewater treatment plants and animal feeding operations have been well characterized, the contribution of these sources to ARG loads at the watershed level is not well understood. In this study, we evaluated the association of fecal pollution, class 1 integron, water quality, and watershed parameters on ARGs distribution in the urban aquatic environment. Currently, Proctor Creek is listed as impaired for fecal coliforms on Georgia’s 303(d) list of impaired waters and the high prevalence of fecal indicators observed in this study supports that designation. Along with the high prevalence of fecal indicators, the results also show a widespread distribution of ARGs across the watershed. In general, moderate to strong positive correlations were observed between ARGs and the HF183 and *E*. *coli* markers, which suggest a strong link between the levels of human fecal pollution and ARG abundance and distribution at the watershed level. Ahmed *et al*. [21] observed a similar association between ARG abundance in storm drain outfalls in Tampa, FL and the levels of sewage-associated HF183 and *crAssphage* markers. 

The *int1* gene, on the contrary, showed a strong positive correlation with the *sulI* gene and a moderate positive correlation with *sulII*, whilst the relationships with the other ARGs were weak. The association of the *int1* gene with the sulfonamide genes, particularly the *sulI* gene, has been well described in previous studies and generally reflects the selective pressure imposed by anthropogenic pollution [37,38]. The observed influence of the sewage-associated HF183 and the *integrase* gene on the *sulI* gene suggests that human fecal pollution and class 1 integrons, respectively, are important drivers of sulfonamide resistance genes in Proctor Creek. Interestingly, a weak but positive correlation was observed between the *int1* gene and the sewage-associated HF183 marker. The occurrence of class 1 integrons from other sources in the environment in addition to human fecal inputs may have contributed to the weak relationship observed between HF183 and *int1* gene [2]. Additionally, the abundance of the 16S rRNA gene showed moderate positive correlations with the sewage-associated HF183 and ARGs, suggesting that increased microbial populations resulting from human fecal inputs are an important driver of ARGs in the urban aquatic environment. 

To identify the important drivers of ARGs in the Proctor Creek watershed, generalized boosting models were developed to predict the average influence of water quality variables and watershed characteristics on ARG concentrations in the watershed. Generalized boosting models are powerful algorithms that make no distributional assumptions on the model error terms, use stochastic methods to improve model performance, and are robust to missing values and outliers [39]. Results from the refined boosted models show that the abundance of ARGs in the watershed could be primarily explained by the concentrations of *E*. *coli* marker, HF183, and the differences between site locations in the watershed. Similar to the sewage-associated HF183 marker, the *E*. *coli* marker has been used in water quality studies as an indicator of fecal pollution due to the high levels of *E*. *coli* in the feces of humans and other warm-blooded animals [40]. The levels of HF183 and *E*. *coli* marker, when considered together, are a strong indicator of the extent of fecal contamination; the model results support the significant contribution of fecal pollution to the abundance of multiple ARG classes in the urban aquatic environment. 

The influence of site is also significant, as the location in the watershed reflects a gradient of human impact from the highly urbanized areas surrounding the headwaters to the low-density residential suburbs around the lower reaches of the watershed. In general, there was a marked increase in ARGs in the highly urbanized areas of the watershed at GR, LST, NAVE, HT, and NAVECSO compared to the lower reaches (NW, LC, JJ, KC, and SST). The combined influence of the fecal indicators and site location on ARG variability in the watershed suggests the significant contribution of sewage/wastewater to ARG loading in the Proctor Creek watershed. Other parameters such as the class 1 integron gene, water temperature, and the bacterial community also contributed to the variability in ARGs at the watershed level, but their effect was insignificant compared to fecal pollution. Additionally, water quality parameters such as turbidity, temperature, antecedent rainfall, and season did not appear to influence ARG concentrations at the watershed level. These findings are in line with a recent metadata study that found that the presence of resistance genes in the environment can largely be explained by fecal pollution, except in environments polluted by very high levels of antibiotics, where selection pressure dominates [22]. As mentioned earlier, the Proctor Creek watershed has a long legacy of fecal pollution from CSOs and sanitary sewer leaks that continues to impact water quality. Efforts have been made to control fecal loading in the watershed by decommissioning a CSO at the GR location and limiting the discharges from the NAVE CSO. However, our results suggest that the GR location is still a hotspot for fecal pollution resulting in increased ARG loading into streams. Further studies are needed in the watershed to assess the sources of fecal pollution at the GR location and to implement the necessary mitigation measures to reduce the impact of fecal pollution on the dissemination of ARGs in streams. Additionally, future work should focus on the contribution of fecal pollution to the spread of antibiotic-resistant bacteria and pathogens in the streams to enable a thorough assessment of the risk to public health. 

## 5. Conclusions

Understanding the contribution of different sources to the spread of ARGs in the aquatic environment is an important first step in curtailing the dissemination of resistance genes in the environment. In this study, we identified the influence of fecal pollution and quantified the contribution of sewage-polluted environments to the abundance and spread of ARGs in urban streams. Our results confirm that the abundance and distribution of ARGs in the Proctor Creek watershed are largely explained by the levels of human fecal pollution in this stream. Other factors, such as the levels of the class 1 integron gene, microbial community abundance, and water temperature, played a lesser role in explaining the variability of ARGs in this stream. The evidence from this study supports the theory that in aquatic environments highly impacted by sewage pollution, the input from fecal sources is a more critical ARG source/driver than other factors such as environmental parameters (i.e., temperature or seasonal patterns) and watershed stressors. Finally, the results from this study provide local watershed managers and stakeholders with the information needed to reduce the burden of ARGs and fecal bacteria in urban streams. Future research should examine the relationships between fecal pollution, ARB, and pathogens to identify specific health risks associated with the presence of ARGs in the aquatic environment. 

## Figures and Tables

**Figure 1 microorganisms-10-01804-f001:**
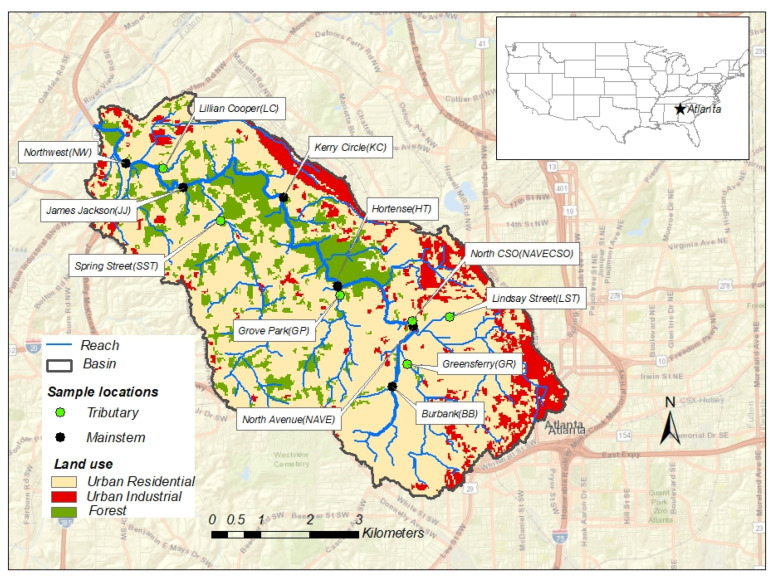
Proctor Creek watershed showing sampling points and land-use distribution.

**Figure 2 microorganisms-10-01804-f002:**
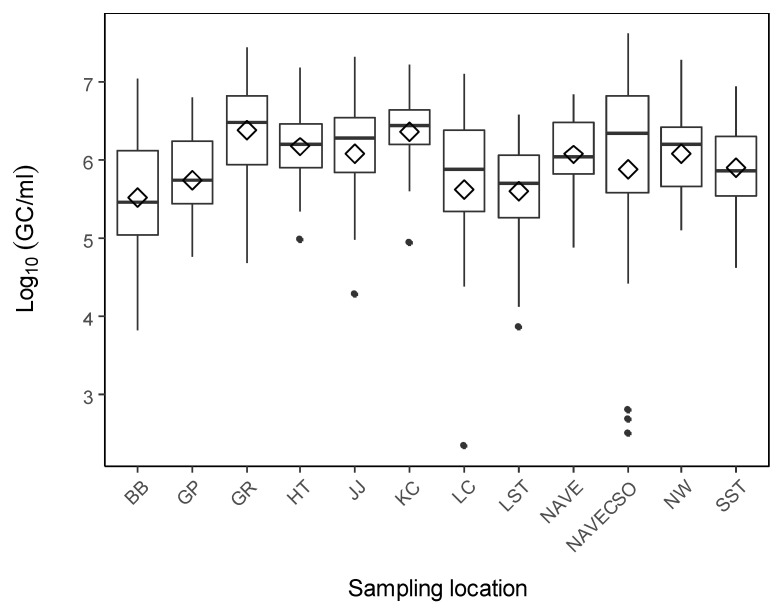
Abundance of 16S rRNA gene copies in surface water samples across the sampling points. The boxplots show the 25th quartile, 75th quartile, and median lines. The diamond shape represents the mean observations for each site, whilst outliers are represented by the solid points.

**Figure 3 microorganisms-10-01804-f003:**
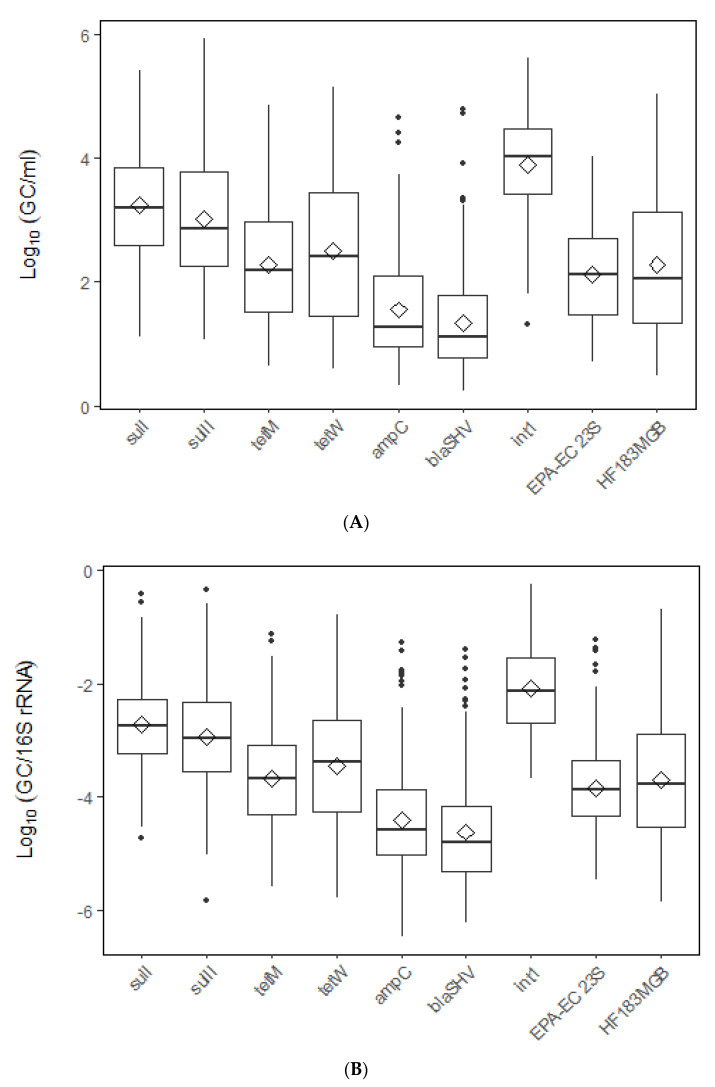
Concentration (**A**) and relative abundance (**B**) of ARGs, fecal markers, and *int1* gene for the pooled dataset. Relative abundance was calculated as the concentration of ARGs normalized to the concentration of the 16S rRNA gene.

**Figure 4 microorganisms-10-01804-f004:**
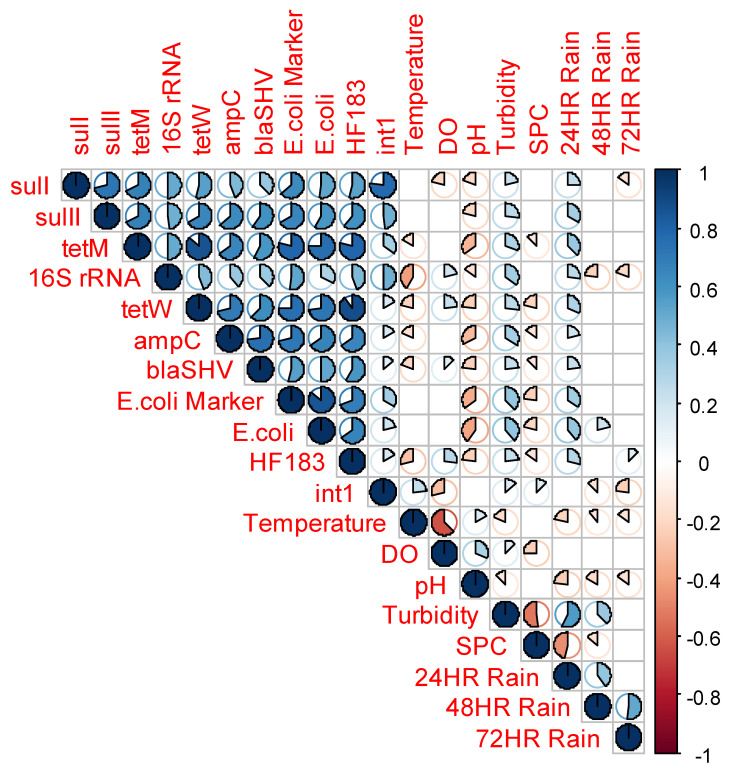
Spearman rank correlations among ARGs, fecal markers, *int1* gene, 16S rRNA, standard water quality parameters, and antecedent rainfall. The correlation *r* is represented by the area of the pie.

**Figure 5 microorganisms-10-01804-f005:**
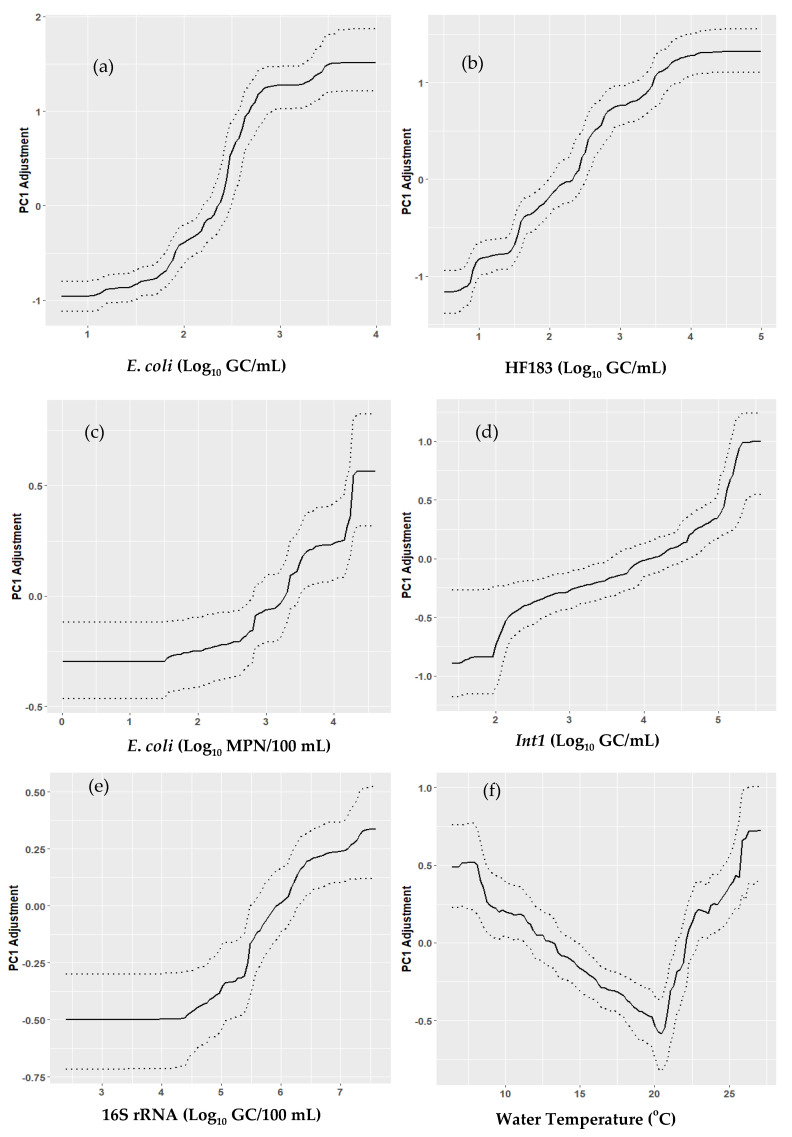
Partial dependence plots showing the response of PC1 scores to changes in predictor variables *E. coli* marker (**a**), HF183 (**b**), culturable *E. coli* (**c**), *int*1 (**d**), 16S (**e**) and water temperature (**f**) in the Refined PC1 model. Dotted line shows the 95% confidence band based on 500 runs of the Refined PC1 model.

**Figure 6 microorganisms-10-01804-f006:**
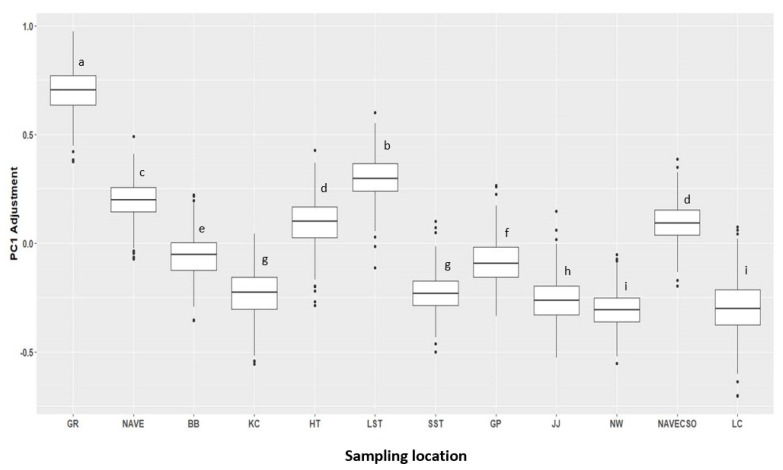
Partial Dependence Plot showing the influence of sampling locations on ARG abundance, as represented on the y-axis by the PC1 scores. Sampling locations are arranged following a gradient of high to low human fecal input from GR to LC. Different letters on boxplots represent significant differences whilst similar letters show no differences.

**Table 1 microorganisms-10-01804-t001:** Detection frequencies (%) of ARGs, class 1 integron, and fecal markers.

Site	*sulI*	*sulII*	*tet*M	*tet*W	*ampC*	*bla*SHV	*int*1	*E. coli*	HF183
Burbank (BB)	96	92	100	100	92	56	96	96	96
Greensferry (GR)	100	96	100	100	100	96	100	96	100
North Ave. (NAVE)	100	100	100	100	84	84	100	100	100
Lindsay Street (LST)	100	100	100	96	40	36	100	76	80
North Ave. CSO (NAVECSO)	96	88	79	46	38	33	96	67	38
Grove Park (GP)	100	96	88	96	24	20	100	92	60
Hortense (HT)	100	100	84	76	40	44	100	92	64
Kerry Circle (KC)	100	100	84	72	72	44	100	100	68
Spring St. (SST)	100	100	80	76	36	32	100	88	56
James Jackson (JJ)	100	96	76	68	40	32	100	88	40
Lillian Cooper (LC)	71	71	79	50	36	21	86	71	0
Northwest (NW)	100	96	72	52	40	40	100	84	52
Mean	97	95	87	78	53	45	98	87	63

**Table 2 microorganisms-10-01804-t002:** Model performance for training and testing data for each of the modeled datasets, with standard deviations in parentheses.

Model	R^2^ Training	R^2^ Testing
PC1	0.99 (0.01)	0.86 (0.03)
Refined PC1	0.95 (0.01)	0.85 (0.04)
PC2	0.99 (0.02)	0.49 (0.10)

**Table 3 microorganisms-10-01804-t003:** Average influence of the predictor variables for the PC1, Refined PC1, and PC2 models based on 500 GBM runs for each. Standard deviations are in parentheses. The 7 predictors above the dotted line were retained and included in the Refined PC1 model.

Parameter	Ave Influence, PC1 (%)	Constraint	Ave Influence, Refined PC1 (%)	Ave Influence, PC2 (%)
*E*. *coli* Marker	34.0 (3.8)	+	36.8 (3.7)	7.9 (1.3)
HF183	27.4 (3.5)	+	29.5 (3.5)	6.3 (0.8)
Site	11.0 (1.0)	None	12.1 (1.2)	15.2 (1.0)
*int*1	6.9 (1.1)	+	6.7 (1.1)	21.7 (2.0)
Culturable *E*. *coli*	5.1 (1.3)	+	5.4 (1.6)	6.5 (1.0)
Water Temperature	3.2 (0.4)	None	6.2 (0.8)	8.3 (0.9)
16S	3.0 (0.4)	+	3.2 (0.6)	3.5 (0.5)
Season	2.2 (0.3)			3.5 (0.6)
Turbidity	1.7 (0.3)			4.4 (0.6)
Dissolved Oxygen	1.5 (0.2)			5.1 (1.0)
pH	1.1 (0.2)			2.8 (0.5)
Specific Conductance	1.0 (0.2)			4.7 (0.7)
48 h Rainfall	0.9 (0.2)			5.3 (1.3)
24 h Rainfall	0.6 (0.1)			3.3 (0.6)
72 h Rainfall	0.6 (0.1)			1.4 (0.4)

^+^ Response variable constrained to be monotonically increasing.

## Data Availability

Data supporting this study can be found at: https://catalog.data.gov/dataset/epa-sciencehub (accessed on 31 August 2022).

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
