# Peer review of "Sources and Drivers of ARGs in Urban Streams in Atlanta, Georgia, USA"

_microorganisms, 2022, doi:10.3390/microorganisms10091804_

Round 1
Reviewer 1 Report
This manuscript reports evaluated the influence of fecal pollution and quantified the contribution of anthropogenic impact to the abundance and spread of ARGs in urban stream, in the City of Atlanta, Georgia, USA. The findings of this paper will be important for the understanding of drivers of ARGs in an urban aquatic environment. This is a will written, interesting, and useful contribution, which I think is entirely suitable for publication in Microorganisms.
I have the following comments to this manuscript for future publication.
Comments:
1. Title- Since this paper is a study of a river in Georgia, USA, it is advisable to include the study area in the title.
Author Response
Thank you for your comments. We have revised the title of the manuscript as follows: "Sources and Drivers of Antibiotic Resistance Genes in Urban Streams in Atlanta, Georgia, USA"
Reviewer 2 Report
Dear editor and authors,
I carefully read the MS entitled "Sources and drivers of ARGs in an urban aquatic environment". The study is interesting and the MS is well written. The only suggestion I have is to check once again the text format, for example the numbers in the tabel 1, the references 1, 9 and 16.
Author Response
The numbers in Table 1 have been formatted to the same style as the rest of the manuscript. The line “NAVESCO” cannot be reformatted to a narrower line due to the size of the site.
References 1, 9, and 16 have been corrected.
Reviewer 3 Report
The manuscript entitled ‘Sources and drivers of ARGs in an urban aquatic environment’ presents a very interesting and up-to-date study regarding the sources and drivers of ARGs in the Proctor Creek watershed. In general lines, the manuscript is well-written, and the findings are useful for the scientific community, US authorities and/or other stakeholders. There are only some editorials in the text that have to be corrected, e.g., use a dot vs comma in L 30 (pollution,).
Author Response
Line 30 has been corrected, and the manuscript was reviewed for consistency.